# Photovoltaic Fuzzy Logical Control MPPT Based on Adaptive Genetic Simulated Annealing Algorithm-Optimized BP Neural Network

**Yan Zhang, Ya-Jun Wang \*, Yong Zhang and Tong Yu**

Department of Electronic and Information Engineering, Liaoning University of Technology,
Jinzhou 121001, China; zy_yjq@162.com (Y.Z.); zh_yong1997@163.com (Y.Z.); yutongyytt@163.com (T.Y.)
\* Correspondence: zruan2021@163.com

**Abstract:** The P–U characteristic curve of the photovoltaic (PV) cell is a single peak curve with only one maximum power point (MPP). However, the fluctuation of the irradiance level and ambient temperature will cause the drift of MPP. In the maximum power point tracking (MPPT) algorithm of PV systems, BP neural network (BPNN) has an unstable learning rate and poor performance, while the genetic algorithm (GA) tends to fall into local optimum. Therefore, a novel PV fuzzy MPPT algorithm based on an adaptive genetic simulated annealing-optimized BP neural network (AGSA-BPNN-FLC) is proposed in this paper. First, the adaptive GA is adopted to generate the corresponding population and increase the population diversity. Second, the simulated annealing (SA) algorithm is applied to the parent and offspring with a higher fitness value to improve the convergence rate of GA, and the optimal weight threshold of BPNN are updated by GA and SA algorithm. Third, the optimized BPNN is employed to predict the MPP voltage of PV cells. Finally, the fuzzy logical control (FLC) is used to eliminate local power oscillation and improve the robustness of the PV system. The proposed algorithm is applied and compared with GA-BPNN, simulated annealing-genetic (SA-GA), particle swarm optimization (PSO), grey wolf optimization (GWO) and FLC algorithm under the condition that both the irradiance and temperature change. Simulation results indicate that the proposed MPPT algorithm is superior to the above-mentioned algorithms with efficiency, steady-state oscillation rate, tracking time and stability accuracy, and they have a good universality and robustness.

**Keywords:** adaptive genetic algorithm; simulated annealing algorithm; artificial neural network; fuzzy logical control; photovoltaic power generation; MPPT



## 1. Introduction

Solar energy is one of the most popular types of renewable clean energy. The total annual installed capacity of PV panels has increased year by year [1]. However, the PV cell exhibits non-linear characteristics, and the extracted output power is significantly affected by irradiance ($G$) level and ambient temperature ($T$) [2]. Therefore, it has been become a research hotspot in the field of PV power generation to improve the efficiency of the PV power system by using corresponding MPPT algorithms [3].

A PV power system consists of a PV cell, power electronics converter and MPPT controller. Currently, the MPPT algorithm for PV systems includes constant voltage tracking (CVT) [4], perturbation and observation (P&O) [5] and an incremental conductance (INC) algorithm [6]. However, the conventional MPPT methods cannot satisfy the actual control requirements under the rapidly changing $G$ and $T$. Therefore, the intelligent MPPT algorithms, such as artificial neural network (ANN) [7], GA [8], FLC, and bacterial foraging algorithm (BFA) [9], have been extensively applied in the PV systems.

Due to the incomplete datasets of $G$ and $T$ in practical projects, ANN reliability has been immensely limited. GA tracks the MPP of power–voltage (P–U) curve via selection,

crossover and mutation progresses, yet the prediction accuracy of GA is low. FLC tracks the MPP by setting a fuzzy rule and membership function, while the FLC has poor dynamic accuracy and tracking performance. BFA has the advantages of parallel search and fast search speed, nevertheless, the chemotaxis and flip processes cannot efficiently handle the datasets of *G* and *T*. Consequently, it is important to propose a universal, simple and valid MPPT algorithm [10].

In [11], Motahhir et al. developed an improved INC method to reduce the power oscillation of a PV cell. The authors of [12] introduced the RBF neural network (RBFNN) to a wind-solar hybrid system, which used a single ANN to extract the MPP of PV cell. In [13], Ferrs ooz Mirza et al. proposed a GA-PID controller to improve the response speed, tracking the characteristic and stabilization accuracy of a PID controller. In [14], Saha et al. presented a PSO–P&O algorithm to maximize the extracted PV power by using the optimal voltage and current of a PV cell. In [15], Zafar et al. proposed the search and rescue algorithm (SRA) to improve the tracking performance and stabilization accuracy of an MPPT controller under partial shading conditions (PSC). In [16], Kraiem et al. developed the P&O–PSO algorithm to boost the performance of a PV system. The authors of [17] presented the tunicate swarm algorithm (TSA) to estimate the optimized value of the unknown parameters of a PV module under standard temperature conditions. An investigation of the above-mentioned literature illustrates that most of the algorithms are interested in the single control algorithm without realizing the mutual optimization.

Given the drawbacks of single MPPT algorithms, such as poor tracking performance and stabilization accuracy, the authors [18] presented a large variation GA–RBFNN algorithm to achieve no error control for microgrid PV system. In [19], Ouahib et al. developed an ANN–PSO algorithm, where the PSO algorithm is used to improve the convergence rate of ANN. Moreover, the optimized ANN is adopted to predict the MPP voltage and power in a day. The authors of [20] developed an adaptive sliding mode control (SMC) and ANN technology to obtain the MPP of a PV cell. In [21], Ozdemir S et al. presented an ANFIS–PSO algorithm to control the zeta chopper converter and achieve zero oscillation tracking. The authors of [22] presented an improved ant colony optimization (ACO) algorithm to improve the tracking characteristic of MPPT controller under PSC. The authors [23] developed the strategy of random reselection of parasitic nests that appeared in the cuckoo search (CS). In addition, the CS and PSO algorithms are employed to optimize the solar PV system model parameters. The authors of [24] combined a modified Aquila Optimizer (AO) with the opposition-based learning (OBL) technique to optimize the adaptive neuro-fuzzy inference system (ANFIS). Among these intelligence algorithms described above, GA has a fast convergence rate and is easy to be combined with other algorithms, and the SA algorithm has strong robustness and universality. This provides additional motivation to develop the intelligent MPPT algorithm in the PV systems.

Although researchers have studied BPNN and FLC extensively, there are few deep sub-optimizations. Since the slow convergence rate of BPNN and the low dynamic accuracy of FLC, a novel PV fuzzy MPPT algorithm based on an adaptive genetic simulated annealing-optimized BPNN (AGSA–BPNN–FLC) is proposed in this paper. First, the parent and offspring populations are generated by adaptive GA (AGA). Second, the SA algorithm is employed to handle the population with higher fitness value, and the AGA and SA algorithm are adopted to optimize the threshold weight of BPNN. Furthermore, the optimized BPNN is employed to predict the MPP voltage of a PV cell ($V_{\text{ref}}$). Finally, the FLC is adopted to eliminate local power oscillation and improve the stability of PV power systems. This paper makes the following contributions: (1) the AGSA–BPNN–FLC algorithm is proposed to improve the tracking characteristics and photoelectric conversion efficiency of PV systems. (2) The proposed algorithm is compared with the GA–BPNN, SA–GA, PSO, GWO and FLC algorithms under rapidly changing *G* and *T*. In addition, Matlab/Simulink is employed to verify the validity and feasibility of the proposed MPPT algorithm.

This paper is organized as follows: a mathematical model of a PV cell and boost converter are introduced in Section 2. Related works including BPNN, FLC, adaptive GA

(AGA), SA and the proposed MPPT algorithm are presented in Section 3. Simulation results and discussions are described in Section 4. The conclusion and future works are presented in Section 5.

## 2. Modelling of PV Power Generation System

### 2.1. Mathematical Model of PV Cell

The PV cell output current equation is given in Equation (1) [25].

$$I = I_{pv} - I_o\left\{\exp\left[\frac{q(U + R_s \times I)}{AKT}\right] - 1\right\} - \frac{U + R_s \times I}{R_{sh}} \tag{1}$$

where: $R_s$ represents the series resistance; $R_{sh}$ is the shunt resistance; $I$ is the output current of PV cell; $U$ is the output voltage of PV cell; $I_{ph}$ is the photo current; $I_o$ is the dark current; $A$ is the ideal parameter of the diode; $K$ is Boltzmann constant ($K = 1.38 \times 10^{-23}$ J/K); $q$ is electron charge ($q = 1.6 \times 10^{-19}$ °C); $T$ is the temperature of PV cell, expressed in °C. Figure 1 illustrates the PV cell equivalent circuit. The I–U–P curves under different $G$ and $T$ are given in Figure 2.

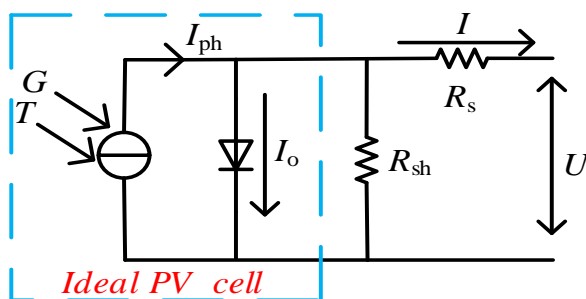

**Figure 1.** PV cell equivalent circuit.

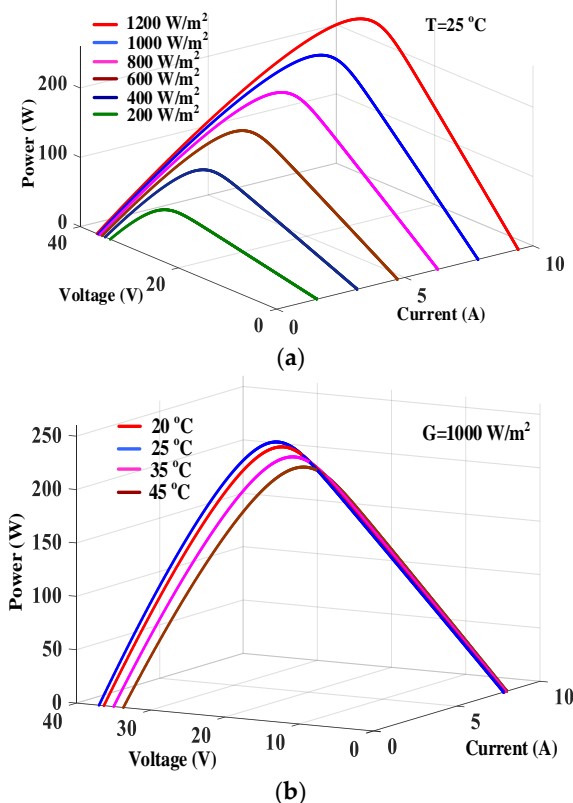

**Figure 2.** PV cell I–U–P curve under: (**a**) different $G$ (**b**) different $T$.

In this paper, the PV cell is selected from the Simulink module library (1Soltecg 1STH-215-P). In Figure 2a, the I–U–P curve is a single peak curve with only one MPP, and the fluctuation of *G* and *T* will cause the drift of MPP. In Figure 2b, the I–U–P curve has strong nonlinearity, and the short-circuit current drastically fluctuates with *G* [26]. Thus, it is vital to propose a sample and effective MPPT algorithm to quickly track the MPP of PV cell. Table 1 shows the PV cell and boost converter parameters.

**Table 1.** PV cell and Boost converter parameters.

| PV Cell Parameters | Value | Boost Converter Parameters | Value |
|---|---|---|---|
| Open circuit voltage $U_{oc}$/V | 37.3 | Capacitor filter $C_1$/μF | 150 |
| Short circuit current $I_{sc}$/A | 8.66 | Boost inductaor $L$/mH | 2.7 |
| MPP voltage $U_{mp}$/V | 30.7 | Output resistant $R$/Ω | 25 |
| MPP current $I_{mp}$/A | 8.15 | Output capacitor $C_2$/μF | 150 |
| $U_{oc}$ temperature coefficient/(%/deg.c) | −0.3609 | — | — |
| $I_{sc}$ temperature coefficient/(%/deg.c) | 0.08699 | — | — |
| Maximum power $P_m$/W | 250.205 | — | — |

### 2.2. Mathematical Model of Boost Converter

Currently, DC–DC topological structure includes the buck–boost converter, boost converter, zeta converter, Cuk converter and more. Among these circuits, the boost converter has the advantages of low ripple and high operating efficiency. Therefore, the boost converter is selected as the carrier of the PV system, as given in Figure 3. The parameter of the boost converter is shown in Table 1. To prevent the PV cell from ending in a mismatched state, the capacitor filter $C_1$ and MPPT algorithm are employed to adjust the load transfer function and state space in real time and reduce energy losses. In addition, the inductive current $I_L$ does not fluctuate with the $G$ and $T$ at steady-state conditions, i.e., $I_{pv} = I_L$. The state equation of the boost converter is given in Equation (2).

$$\begin{cases} dI_{pv}/dt = (U_{pv} - U_o)/L + \alpha U_o/L + \varepsilon \\ dU_o/dt = -U_o/(RC) + (1 - \alpha)I_{pv}/C + \varepsilon \end{cases} \tag{2}$$

where: $I_L$ represents the inductor current; $C_1$ is the capacitor filter; $V_{pv}$ is the PV cell voltage; $I_{pv}$ is the PV cell current; $L$ is the boost inductor; $V_o$ is the output voltage of boost converter; $I_o$ is the output current of the boost converter; $D$ is the duty cycle; $\varepsilon$ is the disturbance quantity.

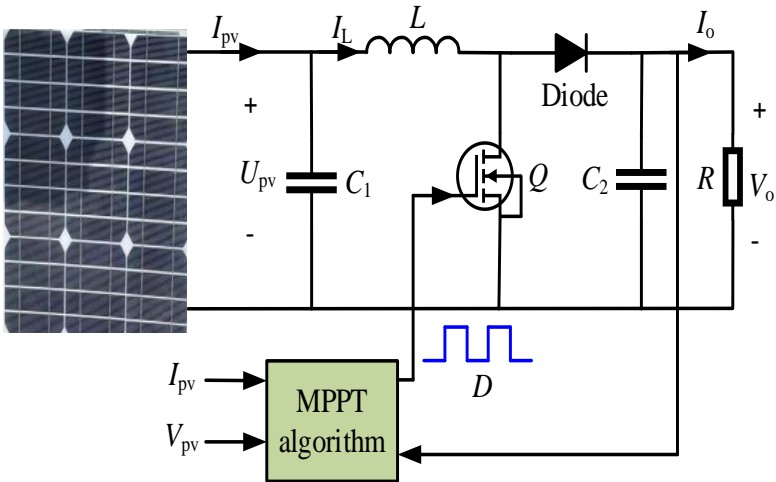

**Figure 3.** Topological of PV Boost converter.

### 3. Related Works

The proposed MPPT algorithm is composed of a PV cell, fuzzy logical control (FLC), genetically simulated annealing-optimized BPNN algorithm and boost converter, as shown in Figure 4. The specific processes are as follows: First, the datasets of $G$ and $T$ are sampled as the inputs of the PV system, meanwhile, the adaptive GA and SA algorithm are used to update the optimal weight threshold of BPNN. Second, the optimized BPNN is adopted to predict the MPP voltage of PV cell ($V_{ref}$). Third, the voltage deviation $\Delta V = V_{pv} - V_{ref}$ and duty ratio $D$ ($n-1$) at time $n-1$ are employed as the inputs of FLC for fuzzification, fuzzy reasoning and defuzzification to control the duty cycle $D$ ($n$). Finally, the PWM is employed to control the on–off time of Mosfet.

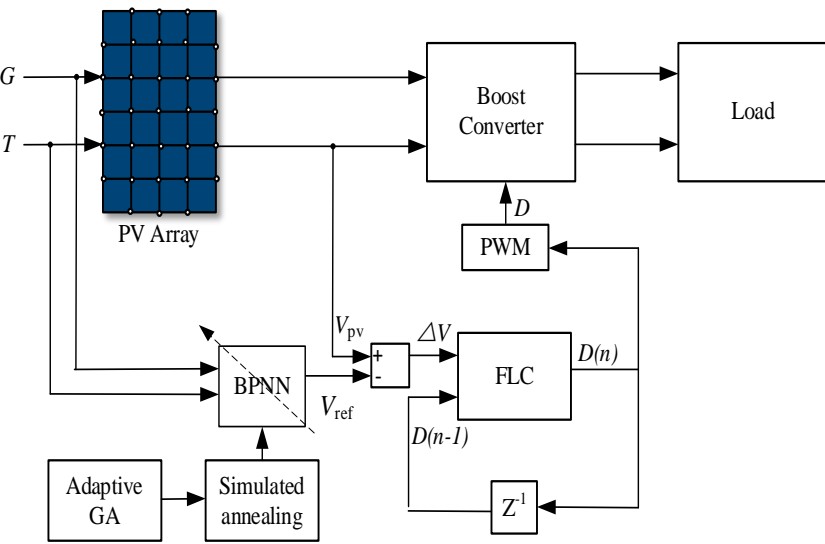

**Figure 4.** Schematic diagram of the proposed algorithm.

### 3.1. Artificial Neural Network (ANN)

ANN is a computing model, which is interconnected by a large number of neurons. BPNN is a global approximation network, having the advantages of simplicity, self-learning and self-adaptation [27]. Figure 5 shows a typical three-layer ANN basic structure.

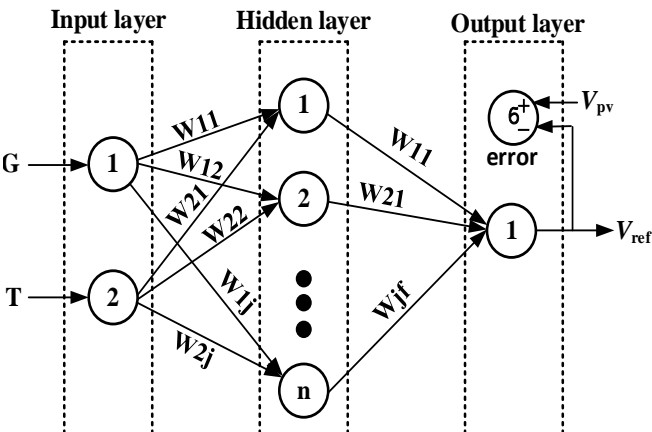

**Figure 5.** ANN basic structure.

As shown in Figure 5, $W_{ij}$ represents the weight of the input layer to the hidden layer; $W_{jf}$ is the weight of the hidden layer to the output layer; $V_{pv}$ is the MPP voltage of the PV cell; $V_{ref}$ is the reference voltage of the PV cell. The number of hidden layer nodes has a crucial impact on the performance of ANN in the forward propagation. If there are fewer nodes, it is impossible to learn and store a majority number of mapping complexes;

otherwise, it will lead to a long learning time and slow convergence rate. Therefore, the number of hidden layer nodes is updated according to Equation (3). The commonly used activation function mainly, including *Sigmoid*, *Purelin*, *Tansig*, *Logsig* [28]. The *Tansig* function can effectively reduce the loss degree of the output layer and normalize to [−1, 1]. Therefore, the *Tansig* is selected as the activation function in this paper. The Levenberg–Marquardt (*L–M*) algorithm is adopted as the training function of BPNN. Equation (4) is the updated formula of learning rate $\eta$. Figure 6 illustrates the MPP voltage of PV cell under varying $G$ and $T$.

$$hiddennum = \sqrt{m + n} + a \tag{3}$$

$$\eta(k) = \begin{cases} a & c_1(k) < 0 \\ -b\eta(k-1) & c_1(k) > 0 \\ 0 & else \end{cases} \tag{4}$$

where: $m$ is the number of input nodes; $n$ is the number of output nodes; $a$ is an integer within [1, 20]; $c_1(k)$ is the square error gradient value of the previous $i$ iterations of the BPNN; $\eta$ is learning rate.

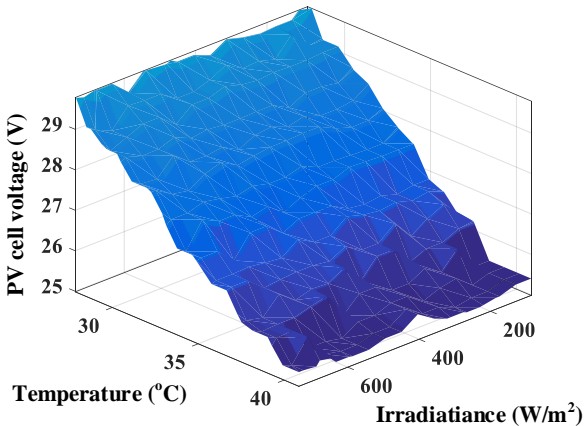

**Figure 6.** MPP voltage of PV cell under varying $G$ and $T$.

### 3.2. Fuzzy Logical Control (FLC)

To solve the oscillation problem of the nonlinear system, the FLC is used in the PV systems. The voltage deviation $\Delta V = V_{pv} - V_{ref}$ and $D (n - 1)$ are taken as the inputs of FLC to improve the optimization time and stabilization accuracy of FLC. Figure 7 shows the structure of FLC [29].

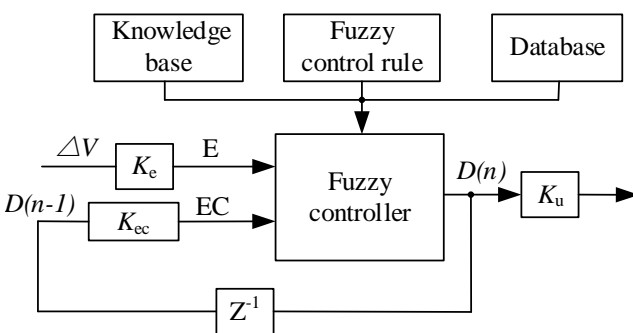

**Figure 7.** The structure of FLC.

In Figure 7, E, EC, D($n$) ∈ NB, NM, NS, Z0, PS, PM, PB}, NB is negative big; NM is negative middle; NS is negative small; Z0 is zero; PS is positive small; PM is positive middle; and PB is positive big. The universe of discourse is mapped as [−1, 1]. $K_e$ and $K_{ec}$ are the inputs quantization factors, and $K_u$ is the output scale factor. The simulation results indicate that the chattering of output waveform is at its minimum when $K_{ec} \approx 6K_e$.

Therefore, $K_e$, $K_{ec}$ and $K_u$ are 1/280, 1/50 and 1/30, respectively. Figure 8 shows the membership function of FLC. Table 2 is the FLC rule.

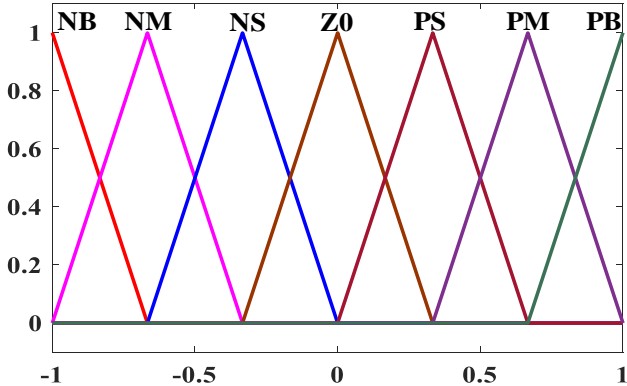

**Figure 8.** The membership function of FLC.

**Table 2.** FLC rule.

| E/EC | NB | NM | NS | Z0 | PS | PM | PB |
|------|-----|-----|-----|-----|-----|-----|-----|
| NB | NB | NB | NB | NM | NM | NM | NB |
| NM | NB | NB | NM | NS | NS | NS | NM |
| NS | NB | NS | NS | Z0 | NS | NS | NM |
| Z0 | NS | NS | NS | Z0 | PS | PS | PS |
| PS | PM | PS | PS | PS | PM | PB | PB |
| PM | PB | PM | PS | PS | PM | PB | PB |
| PB | PB | PB | PB | PM | PS | PM | PB |

### 3.3. Adaptive GA and SA Algorithm

GA is a global search algorithm that automatically obtains the search direction of the parent population in the search process and finds the optimal solution by evaluating the fitness value [30]. The tournament, roulette and elite selection strategy are commonly used for the selection of individuals, however, the optimization effect is not obvious and the complexity is visibly increased.

The crossover and mutation probabilities determine the population diversity and convergence rate. In the conventional GA, the crossover probability ($P_c$) and mutation probability ($P_m$) are constants. Therefore, the conventional GA has difficultly quickly tracking the MPP of the PV cell. Therefore, this paper proposes an adaptive crossover and mutation strategy.

There are many optimization parameters in the PV system. Hence, the relationship between maximum fitness value ($f_{max}$) and average fitness value ($f_{avg}$), i.e., $f_{max} - f_{avg}$, is used to judge the concentration degree of population and dynamically adjust the diversity and randomness of population. For individuals with low fitness value, $P_m$ and $P_c$ are appropriately increased; otherwise, $P_m$ and $P_c$ are reduced. The adaptive crossover and mutation probabilities are defined in Equations (5)–(7), which can avoid the local optimal solution and improve the convergence rate.

$$P_c = \begin{cases} k_1 \frac{f_{max} - f'}{f_{max} - f_{avg}} , & f' \geq f_{avg} \\ k_3 , & f' \leq f_{avg} \end{cases} \tag{5}$$

$$P_m = \begin{cases} k_2 \frac{f_{max} - f'}{f_{max} - f_{avg}} , & f' \geq f_{avg} \\ k_4 , & f' \leq f_{avg} \end{cases} \tag{6}$$

$$\begin{cases} p_c = k_3, & f' \le f_{avg} \\ p_m = k_4, & f \le f_{avg} \\ 0.4 \le k_3, & k_4 \le 1 \end{cases} \tag{7}$$

where: $f'$ represents the larger fitness value of the two crossover individuals; $f$ is the fitness value of mutated individual; $f_{max}$ is the maximum fitness value of the population; $f_{avg}$ is the average fitness value of the population; $k_3$ is the minimum crossover probability value; $k_4$ is the minimum mutation probability value; $k_1$ and $k_2$ are the maximum crossover probability coefficient and the maximum variation probability coefficient, respectively.

SA is a greedy algorithm based on the annealing of solids in physics, which eliminates the non-uniform state in the physical system by using the processes of heating and isothermal cooling. The heating process is the parameter initialization [31], the isothermal process corresponds to Metropolis sampling, and the cooling process corresponds to the decrease in parameters. Metropolis criterion is the core of SA algorithm, which receives deteriorating solutions with a certain probability $p$. The climbing speed and global searching speed of GA are enhanced by performing annealing operations on individuals with higher fitness value. Metropolis criterion is given in Equation (8). Annealing methods mainly include linear cooling, exponential cooling and logarithmic cooling. Exponential annealing is selected in this paper, which is given in Equation (9).

$$p = \begin{cases} 1, & E_{t+1} < E_t \\ e^{-\frac{(E_{t+1} - E_t)}{KT}}, & E_{t+1} \ge E_t \end{cases} \tag{8}$$

$$T_j = T_0 \times k^{j-1} \tag{9}$$

where: $T_j$ is the current annealing temperature; $T_0$ is the initial annealing temperature; $k$ is the temperature attenuation coefficient; $E_{t+1}$ and $E_t$ are the energies at moments $t$ and $t + 1$, respectively; $K$ is the Boltzmann constant; $p$ is the probability of acceptance.

### 3.4. AGSA–BPNN–FLC Algorithm

Figure 9 shows the proposed AGSA–BPNN–FLC algorithm flowchart. The main steps are as follows:

1.  Confirm the BPNN structure and initialize parameters, such as population size $nPop$, maximum evolutionary generation $It_{max}$, crossover probability $P_c$, variation probability $P_m$, initial annealing temperature $T_0$, temperature attenuation coefficient $k$, maximum annealing number $T\_It$ and Markov chain length $L$.
2.  The selection, crossover and mutation operations are carried out to generate the offspring populations, where the crossover and mutation probability are calculated according to Equations (5)–(7).
3.  The SA algorithm is adopted for individuals with high fitness value. According to the Equations (8) and (9), the deterioration solution is received with a certain probability $p$. If the real-time probability is greater than the reception probability $p$, the new solution is completely received; otherwise, the new solution is received with probability $p$.
4.  Determine whether the current temperature $T$ is less than the final temperature $T_{min}$. If $T < T_{min}$, the algorithm ends; otherwise, return to step (3) to execute the SA algorithm.
5.  Judge whether the current iteration number $It$ satisfies the maximum evolution algebra $It_{max}$. If $It < It_{max}$, then $It = It + 1$; otherwise, return to step (2).
6.  The optimized BPNN is used to predict the MPP voltage ($V_{ref}$).
7.  The voltage deviation $\Delta V = V_{pv} - V_{ref}$ and $D(n-1)$ are employed as the inputs of FLC to obtain the duty ratio $D(n)$ of the boost converter. Figure 10 shows the best performance curve of the optimized BPNN.

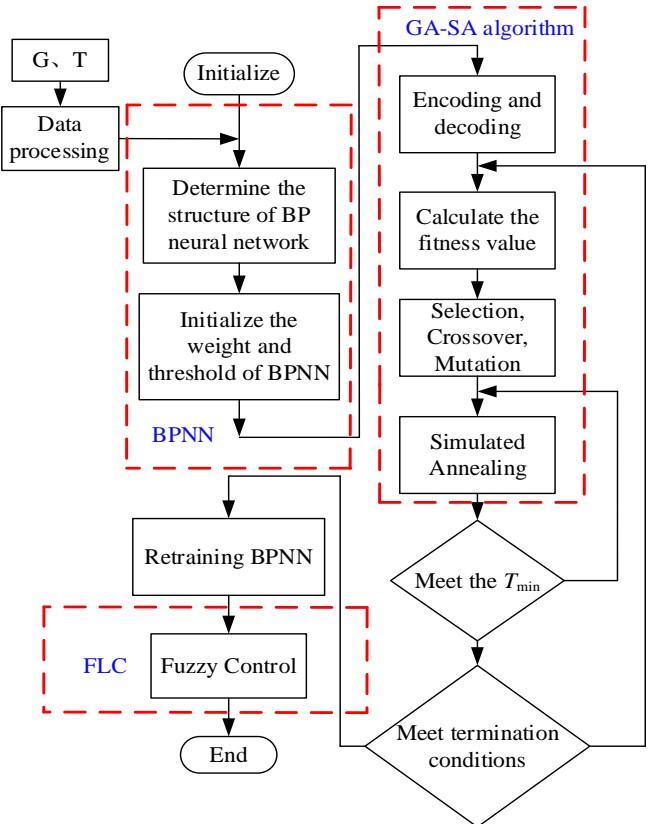

**Figure 9.** The proposed AGSA–BPNN–FLC algorithm flowchart.

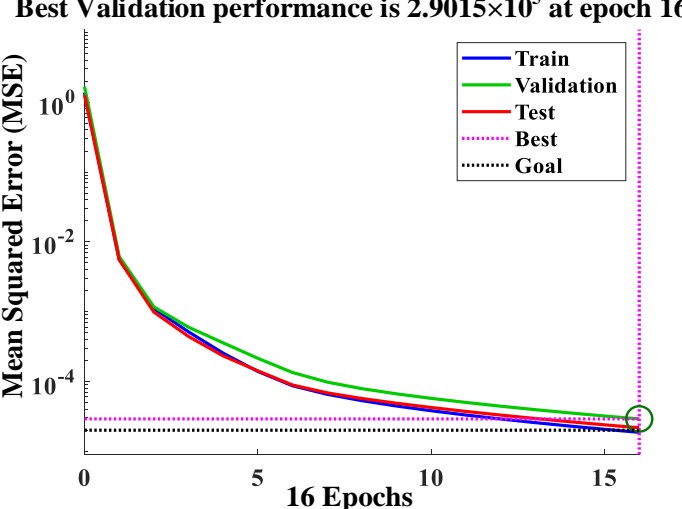

**Figure 10.** The best performance curve of optimized BPNN.

As shown in Figure 10, the Epochs and MSE value of optimized BPNN are 16 and $2.9015 \times 10^{-5}$, respectively. Simulation results indicate that the optimization effect and convergence rate are better when population size (*nPop*) is about two times of the maximum number of iterations (*It*$_{max}$), i.e., $nPop \approx 2It_{max}$. In addition, the temperature attenuation coefficient (*k*) and initial temperature ($T_0$) have a strong influence on the convergence rate of the optimized BPNN. Therefore, the parameters of the proposed MPPT algorithm are given in Table 3. The advantages and disadvantages of the GA–SA and SA–GA algorithms are discussed and analyzed in Section 4.2.

**Table 3.** Proposed algorithm parameter.

| Parameters | Value |
| --- | --- |
| Hiddennnum | 30 |
| Epochs | 1000 |
| Population size *nPop* | 50 |
| Maximum number of iterations $It_{max}$ | 100 |
| Initial temperature $T_0$ | 100 |
| Markov chain *L* | 10 |
| Temperature attenuation coefficient *k* | 0.85 |
| Maximum annealing times $T\_It$ | 10 |

## 4. Simulation Results and Discussion

### 4.1. Datasets and Simulation Model

Matlab is used to obtain the inputs and output datasets, the inputs are the irradiance level and ambient temperature and the output is the MPP voltage ($V_{mpp}$). The inputs and output datasets are obtained by 10,000 cycles of Equations (10)–(12). A total of 70% of the datasets are the training sets and the rest is the testing sets.

$$G = (G_{max} - G_{min}) \times rand + G_{min} \tag{10}$$

$$T = (T_{max} - T_{min}) \times rand + T_{min} \tag{11}$$

$$V_{mpp} = V_{mps} + (beta \times (T - T_{ref}) \tag{12}$$

where: $G_{max}$ is 1000 W/m$^2$; $G_{min}$ = 0 W/m$^2$; $T_{max}$ = 40 °C; $T_{min}$ = 10 °C; *rand* is a random value in [1]; $V_{mps}$ is the MPP voltage of PV cell under standard test conditions (STC, $G_{ref}$ = 1000 W/m$^2$, $T_{ref}$ = 25 °C); *beta* is the temperature coefficient.

The boost converter of the PV system is built under Simulink, as given in Figure 11. As shown in Figure 11, the proposed algorithm consists of a PV cell, boost converter, optimized BPNN and FLC. The proposed MPPT algorithm is compared with GA–BPNN, SA–GA, PSO, GWO and FLC algorithms to verify the feasibility and effectiveness. The simulation time is 1.5 s, and the simulation algorithm is fixed-step (ode3).

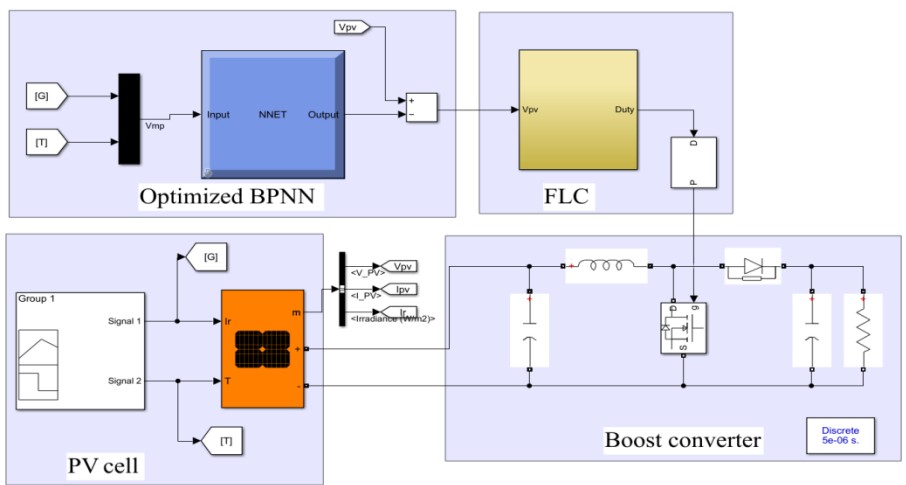

**Figure 11.** PV system under AGSA–BPNN–FLC control.

### 4.2. Optimized Results

In this paper, two optimization algorithms, GA–SA and SA–GA algorithms, are compared and studied. SA–GA algorithm uses GA to initialize the population and then performs SA algorithm until the cooling schedule is satisfied. However, the SA algorithm is not optimized by the GA, the prediction accuracy is poor. The GA–SA algorithm generates the parent and offspring populations via selection, crossover and mutation operations. The

parent and offspring populations with higher fitness value are annealed and the optimal parent and offspring are selected as the initial populations for the next iteration until the termination condition is reached. The GA–SA algorithm combines the advantages of wide global search ability of GA and strong local search ability of SA, which improves the climbing speed of GA and the operating efficiency of SA. Equation (13) is the loss function. Figure 12 shows the prediction error of the GA–SA and SA–GA algorithms.

$$RMSE = \sqrt{\frac{1}{n}\sum_{i=1}^{n}\sum_{j=1}^{m}[Y_j(i) - T_j(i)]^2} \tag{13}$$

where: $n$ represents the input sets; $m$ is the output sets; $Y_j(i)$ is the true value; $T_j(i)$ is the target value.

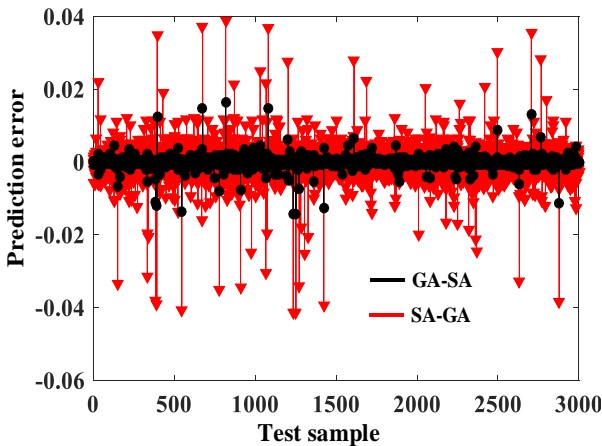

**Figure 12.** The prediction error of GA–SA and SA–GA algorithm.

As shown in Figure 12, the testing sets is selected as the test sample, and the prediction error of SA–GA is within [−0.04, 0.04]. In contrast, the GA–SA is within [−0.005, 0.005]. It is suggested that the prediction accuracy of the optimized BPNN has immensely improved. Figure 13 shows the loss function of the GA–SA, SA–GA, PSO and GWO algorithms.

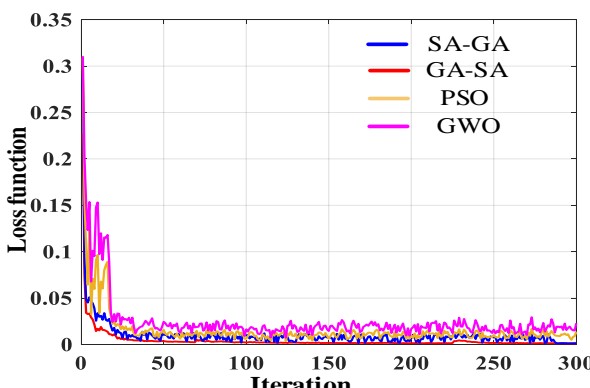

**Figure 13.** The loss function of GA–SA, SA–GA, PSO and GWO algorithm.

As shown in Figure 13, the loss function value of GA–SA is within [0.001, 0.004]. In contrast, the loss function value of SA–GA, PSO and GWO is within [0.002, 0.006], [0.004, 0.008] and [0.006, 0013], respectively. It is clearly seen that the GA–SA algorithm has a good tracking performance, search range and convergence rate. The MSE curves of BPNN, RBF neural network (RBFNN) and optimized BPNN are given in Figure 14.

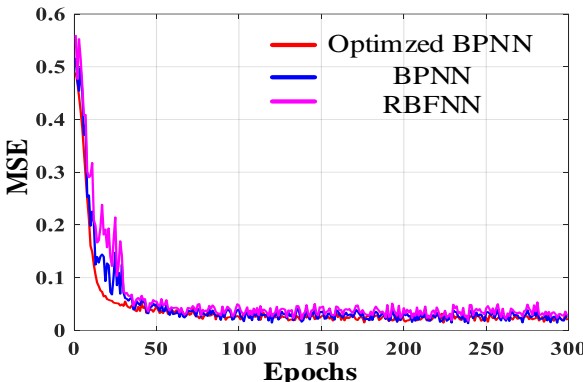

**Figure 14.** The MSE curves of the conventional BPNN, RNFNN, and optimized BPNN.

As shown in Figure 14, the MSE value of optimized BPNN is within [0.02, 0.03]. In contrast, the MSE value of BPNN and RBFNN is within [0.03, 0.05] and [0.04, 0.06], respectively. It is quite clear that the optimized BPNN is superior to the BPNN and RBFNN with convergence rate and reasoning ability in the PV systems.

### 4.3. Simulation Results

The $G$ and $T$ have remarkable influence on the photoelectric conversion efficiency in the practical PV system. The parameters of $G$ and $T$ are as follows: in 0 s~0.25 s, $G = 0$ W/m$^2$, $T = 0$ °C; in 0.25 s~0.5 s (Stage 1), $G = 500$ W/m$^2$, $T = 10$ °C; in 0.5 s~0.75 s (Stage 2), $G = 700$ W/m$^2$, $T = 20$ °C; In 0.75 s~1 s (Stage 3), $G$ decreases from 1200 W/m$^2$ to 1000 W/m$^2$, $T$ decreases from 40 °C to 35 °C; In 1 s~1.25 s (Stage 4), $G = 800$ W/m$^2$, $T = 30$ °C; In 1.25 s~1.5 s, $G = 0$ W/m$^2$, $T = 0$ °C. The proposed MPPT algorithm is compared with the GA–BPNN, SA-GA, PSO, GWO and FLC algorithms to simulate the voltage, power and efficiency of PV cell and the output voltage, output power and duty cycle of the boost converter, as shown in Figures 15–20.

It can be seen from Figures 15–17 that each index of PV system controlled by AGSA–BPNN–FLC algorithm is superior to the five compared MPPT algorithms. Figure 15 shows the voltage waveform of PV cell.

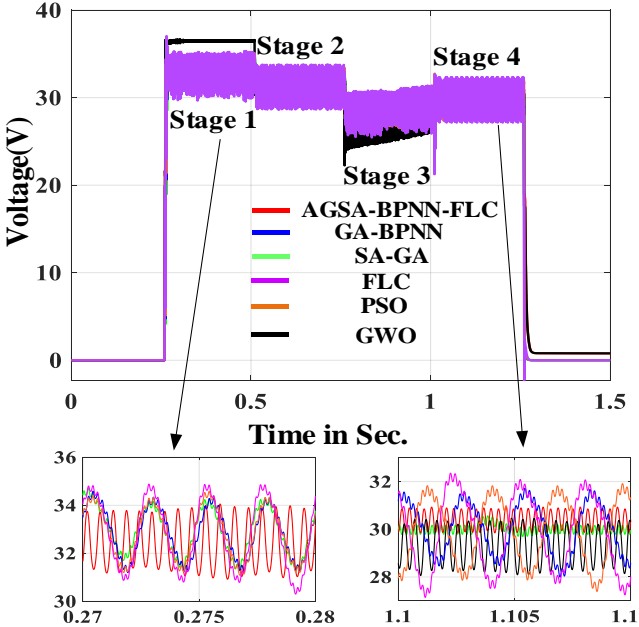

**Figure 15.** The voltage waveform of PV cell.

In Figure 15, the MPP voltage of the AGSA–BPNN–FLC, SA–GA, GA–BPNN, PSO, GWO and FLC algorithms are 33.5 V, 32.8 V, 32.5 V, 31.9 V, 31.3 V and 32.1 V, respectively. It is clearly seen that the proposed MPPT algorithm has a better tracking performance and the waveform is smoother without chattering. In addition, Equations (14) and (15) are formulas for calculating steady-state oscillation rate $\alpha$ and efficiency $\eta$. Figure 16 shows the power waveform of PV cell.

$$\alpha = \frac{P_{smax} - P_{smin}}{P_{smax}} \tag{14}$$

$$\eta = \frac{\sum\limits_{i=1}^{m} P_{out}(m)}{\sum\limits_{i=1}^{m} P_{max}(m)} \tag{15}$$

where: $\alpha$ represents the steady-state oscillation rate; $P_{smax}$ is the steady-state maximum power; $P_{smin}$ is the steady-state minimum power; $P_{out}$ is the output power; $P_{max}$ is the maximum output power; $i$ is the number of samples; $m$ is the sampling time.

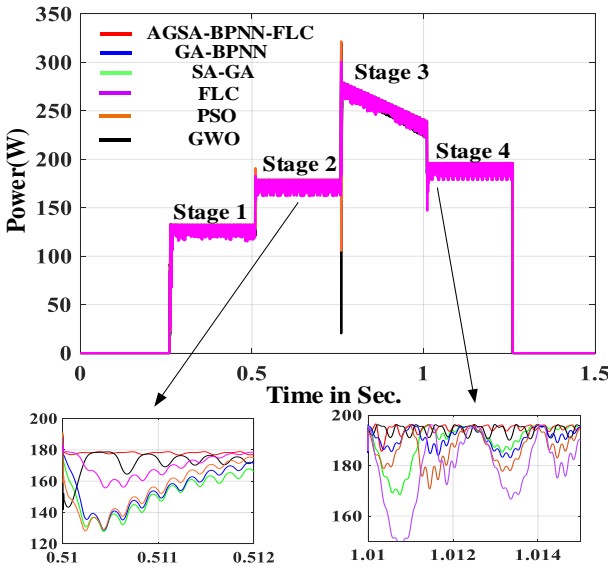

**Figure 16.** The power waveform of PV cell.

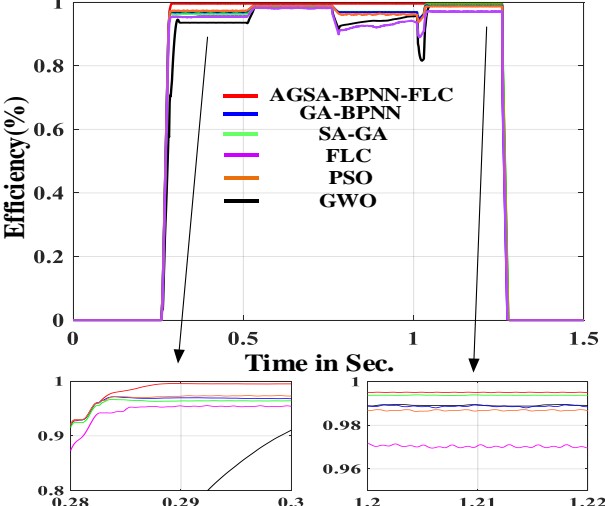

**Figure 17.** Efficiency.

As shown in Figure 16, the steady-state oscillation rates α of proposed MPPT algorithm is 0.3 %. In contrast, the steady-state oscillation rates of the SA–GA, GA–BPNN, PSO, GWO and FLC algorithms are 0.62%, 0.63%, 0.83%, 0.95% and 3.8%, respectively. Meanwhile, the tracking time of proposed MPPT algorithm is 0.003 s. It can be seen from Figure 16 that the tracking performance of the five compared algorithms are 0.008 s, 0.009 s, 0.013 s, 0.018 s and 0.021 s, respectively. It is clearly seen that the proposed MPPT algorithm has stronger tracking characteristic and stability. Figure 17 shows the efficiency of PV cell.

In Figure 17, the efficiency of proposed MPPT algorithm is 99.6%. As can be seen, the efficiency of the five compared algorithms are 98.1%, 97.83%, 97.3%, 96.9% and 96.1%, respectively. From an efficiency viewpoint, the proposed MPPT algorithm has stronger adaptability and higher photoelectric conversion efficiency.

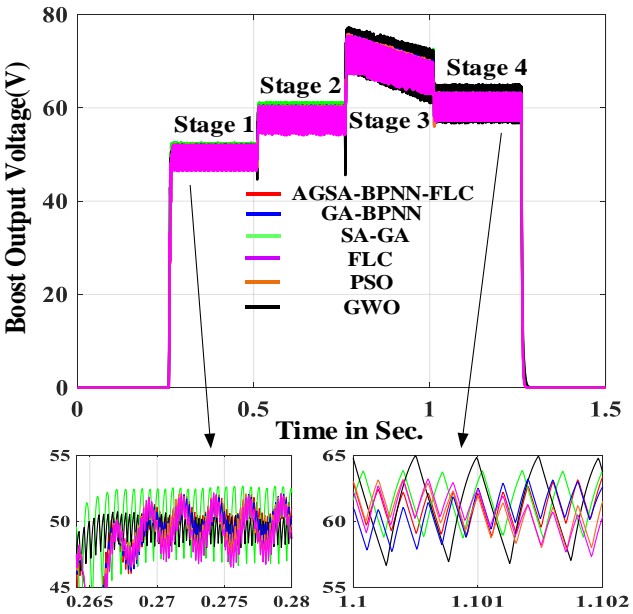

**Figure 18.** Boost converter output voltage.

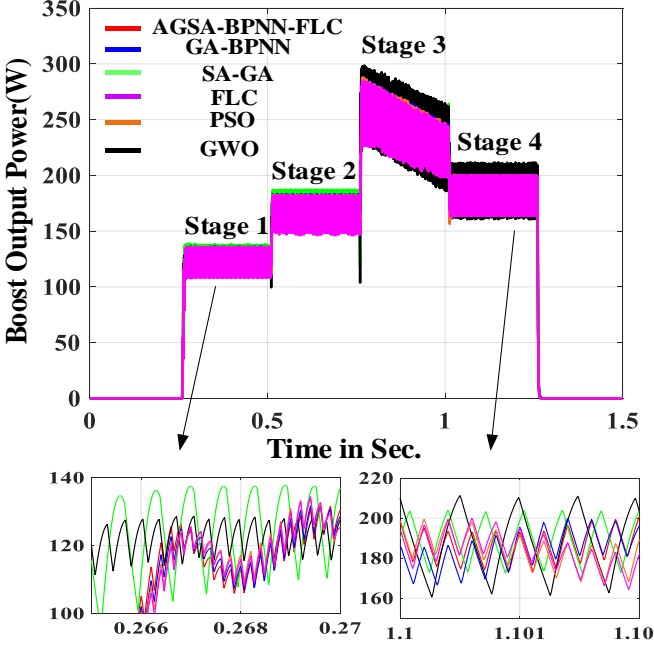

**Figure 19.** Boost converter output power.

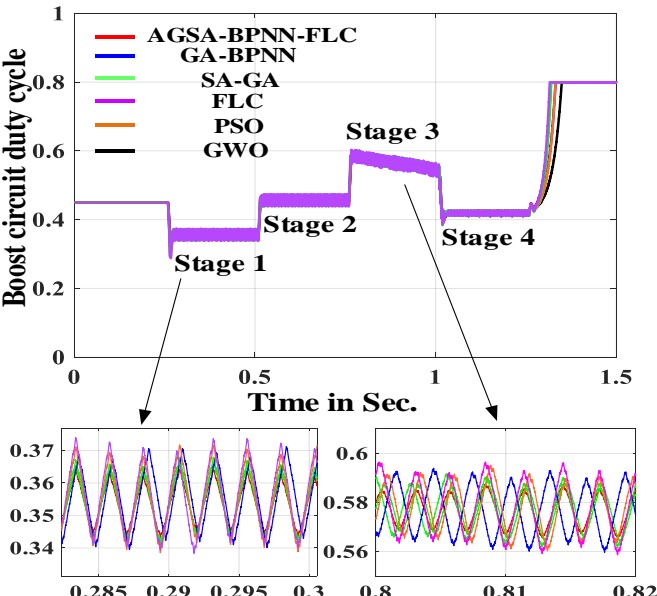

**Figure 20.** Boost converter duty cycle.

In brief, the proposed MPPT algorithm has better tracking characteristics, stabilization accuracy and efficiency than the five compare algorithms.

The output voltage, power and duty cycle waveform of boost converter are given in Figures 18–20. As shown in Figures 18 and 19, the proposed MPPT algorithm is more accurate in determining the optimal range and maximum power boundary point of the P–U curve. Moreover, the proposed MPPT algorithm has a high stability accuracy, good tracking performance and low steady-state oscillation rate. Figure 20 shows the duty cycle of the boost converter.

In Figure 20, the duty cycle of the boost converter is within [0.25, 0.8] to prevent the sub-harmonic oscillation, and the duty cycle can be adjusted more quickly. These obtained results indicate that the tracking characteristic and stabilization accuracy of the proposed MPPT algorithm is superior to the SA–GA, GA–BPNN, FLC, PSO and GWO algorithm. The specific result of each stage are listed in Table 4.

**Table 4.** The specific result of each stage.

| MPPT Algorithm | Amplitude of Power Oscillation/W | | | | Tracking Time/s | | | | Efficiency/% | | | |
|---|---|---|---|---|---|---|---|---|---|---|---|---|
| | Stage 1 | Stage 2 | Stage 3 | Stage 4 | Stage 1 | Stage 2 | Stage 3 | Stage 4 | Stage 1 | Stage 2 | Stage 3 | Stage 4 |
| Proposed algorithm | 0.73 | 0.75 | 0.81 | 0.75 | 0.003 | 0.002 | 0.002 | 0.003 | 99.6 | 99.5 | 99.6 | 99.5 |
| SA–GA | 1.23 | 1.26 | 1.30 | 1.28 | 0.008 | 0.009 | 0.008 | 0.006 | 98.1 | 97.9 | 98.2 | 98.3 |
| GA–BPNN | 1.76 | 1.9 | 1.86 | 1.74 | 0.009 | 0.008 | 0.009 | 0.008 | 97.83 | 97.3 | 97.8 | 97.2 |
| PSO | 1.85 | 1.87 | 1.93 | 1.88 | 0.013 | 0.010 | 0.012 | 0.015 | 97.3 | 97.1 | 97.3 | 97.4 |
| GWO | 2.11 | 2.17 | 2.25 | 2.19 | 0.018 | 0.012 | 0.014 | 0.016 | 96.9 | 96.8 | 97.1 | 96.8 |
| FLC | 4.8 | 5.6 | 5.4 | 4.93 | 0.021 | 0.023 | 0.045 | 0.032 | 96.1 | 96.2 | 96.4 | 96.3 |

| MPPT Algorithm | Steady state oscillation rate (%) | | | | Maximum power deviation/W | | | |
|---|---|---|---|---|---|---|---|---|
| | Stage 1 | Stage 2 | Stage 3 | Stage 4 | Stage 1 | Stage 2 | Stage 3 | Stage 4 |
| Proposed algorithm | 0.3 | 0.38 | 0.41 | 0.35 | 0.07 | 0.11 | 0.13 | 0.09 |
| SA–GA | 0.71 | 0.73 | 0.75 | 0.74 | 0.21 | 0.27 | 0.33 | 0.25 |
| GA–BPNN | 0.93 | 0.91 | 0.96 | 0.89 | 0.34 | 0.41 | 0.45 | 0.37 |
| PSO | 1.13 | 1.17 | 1.21 | 1.15 | 0.64 | 0.67 | 0.71 | 0.65 |
| GWO | 1.25 | 1.27 | 1.32 | 1.23 | 0.71 | 0.75 | 0.82 | 0.73 |
| FLC | 3.5 | 3.8 | 3.7 | 3.5 | 1.2 | 1.4 | 1.5 | 1.3 |

Since the variance test has a high false positive rate and complexity, the Friedman test, Kruskal–Wallis (*K–W*) and *p* value are employed to illustrate the superiority and feasibility of the proposed algorithm. Friedman test is a type of non-parametric test, which is commonly used to compare whether there is a significant difference between two groups. When the Friedman test is adopted to compare each levels, a pair of hypotheses are required,

i.e., null hypothesis ($H_0$) and alternative hypothesis ($H_1$). The null hypothesis indicates that there is a difference, while the alternative hypothesis represents that there is no difference. The hypotheses are as follows:

**H₀.** *there are no significant differences between the proposed algorithm and the five compared algorithms.*

**H₁.** *there are significant differences between the proposed algorithm and the five contrasting algorithms.*

The Friedman test statistic is given in Equation (16). The Friedman test statistic ($p$) is compared with the significance level ($\alpha = 0.05$) to judge the significant differences. If $p < 0.05$, the null hypothesis ($H_0$) is rejected, i.e., the proposed algorithm is superior to the five compared algorithms. However, the Friedman test only focuses on whether there is a significant difference between the columns, ignoring the differences of different data. On this basis, $K$–$W$ and $p$ value are employed to test the population distribution and reliability of the proposed algorithm, respectively. $K$–$W$ judges the significance of each group by calculating the rank and average rank of each data. Moreover, the $K$–$W$ has the characteristics of high efficiency and low complexity. The $p$ value test directly compares the $p$ with the significance level $\alpha$, and the $p$ does not require comparison with the critical value at a given significance level $\alpha$. Therefore, the results of the $p$ value test are more intuitive. Equations (17) and (18) are the $K$–$W$ statistic and $p$ value, respectively.

$$X^2 = \frac{12}{nk(k+1)}\left[\sum R_i^2 - 3n(k+1)\right] \tag{16}$$

$$K - W = \frac{12}{N(N+1)}\sum_{i=1}^{k} ni(\overline{R_i} - \overline{R})^2 \tag{17}$$

$$p = P\left\{F = \frac{(n-k)S_R^2}{kS_E^2} > c\right\} \tag{18}$$

where: $R_i$ represents the average rank; $k$ is the number of sample groups; $N$ is the sample size of group $i$; $\overline{R_i}$ is the average rank of group $i$ ($R_i/n_i$); $\overline{R}$ is the total average rank $[(N+1)/2]$; $F$ is the sample observations; $S_R^2$ is the regression sum of square; $S_E^2$ is the residual sum of square; $P$ is the probability. Equations (16)–(18) have the same hypotheses. Table 5 shows the results of significance test.

**Table 5.** The results of significance test.

| Indicator | Friedman Test | | Kruskal–Wallis Test | | $p$ Value |
|---|---|---|---|---|---|
| | Mean Square | $P$ | Mean Square | $P$ | $P$ |
| Amplitude of power oscillation | 0.4167 | 0.0083 | 3.079 | 0.0003 | 0.0061 |
| Tracking time | 1.5722 | 0.0076 | 3.938 | 0.0006 | 0.0093 |
| Efficiency | 1.1719 | 0.0446 | 3.257 | 0.0007 | 0.0321 |
| Steady state oscillation rate | 0.6556 | 0.0072 | 1.639 | 0.0004 | 0.0083 |
| Maximum power deviation | 0.8652 | 0.0004 | 1.854 | 0.0005 | 0.0045 |

It can be seen from Table 5 that the $p$ value of Friedman test, $K$–$W$ and $p$ value test are less than the significance level ($p < 0.05$). The obtained results indicate that the alternative hypothesis is true, and the proposed algorithm is superior to the five compared algorithms with the amplitude of power oscillation, tracking time, efficiency, steady state oscillation rate and maximum power deviation.

In conclusion, the proposed MPPT algorithm has faster tracking characteristic and stronger stability. At the same time, the proposed algorithm can quickly respond to external condition and has lower buffeting.

### 5. Conclusions and Future Works

This paper proposes a novel adaptive genetic simulated annealing-optimized BP neural network PV fuzzy MPPT algorithm, which can solve the poor tracking performance, stabilization accuracy and low efficiency of PV system. First, $f_{\max} - f_{\text{avg}}$ is introduced to dynamically adjust the crossover and mutation probability. Second, the parent and offspring populations with higher fitness value are simulated and annealed by Metropolis criterion to update the optimal weight threshold of the BPNN. Third, the optimized BPNN is used to predict the MPP voltage of the PV cell. Finally, the voltage deviation $\Delta V = V_{\text{pv}} - V_{\text{ref}}$ and $D\,(n-1)$ are used for fuzzy inference to adjust the on–off time of Mosfet.

The proposed AGSA–BPNN–FLC algorithm is applied to the MPPT technology of the PV power system. The simulation results indicate that the proposed MPPT algorithm can quickly respond to external conditions and has stronger robustness and stabilization accuracy. In addition, the efficiency of proposed MPPT algorithm is improved by 1.5%, 1.77%, 2.3%, 2.7% and 3.5% as compared to the SA–GA, GA–BPNN, PSO, GWO and FLC algorithms. However, it is found that the optimized BPNN cannot be applied to partial shading conditions in the study. Moreover, the prediction accuracy and convergence rate of the optimized BPNN are lower than the deep ANN. Therefore, the future work will adopt the convolutional neural network (CNN) or deep residual neural network (RNN) to improve the prediction accuracy. At the same time, we plan to implement the proposed MPPT algorithm in STM32 or FPGA to achieve MPPT. Moreover, the proposed MPPT algorithm can be applied to solar-powered water pumps and microgrids that improve the efficiency of PV systems.

**Author Contributions:** Y.Z. (Yan Zhang) and Y.-J.W. conceived of the presented idea design and simulation; Y.Z. (Yong Zhang) and Y.-J.W. carried out testing and verification; Y.Z. (Yong Zhang) wrote the original draft of this article; Y.Z. (Yong Zhang), Y.-J.W. and Y.Z. (Yan Zhang) reviewed and edited this article; Y.-J.W. and T.Y. supervised the findings of this work. All authors provided critical feedback and helped shape the research, analysis, and manuscript. All authors have read and agreed to the published version of the manuscript.

**Funding:** This study was supported by the National Natural Science Foundation of China (grant: 61503169, 61802161) and Natural Science Foundation of Liaoning province (grant: 2020-MS-291).

**Institutional Review Board Statement:** Not applicable.

**Informed Consent Statement:** Not applicable.

**Data Availability Statement:** The data presented in this study are not publicly available due to privacy issues.

**Acknowledgments:** The authors would like to thank the National Natural Science Foundation of China and the Natural Science Foundation of Liaoning province for the scholarship.

**Conflicts of Interest:** The authors declare no conflict of interest.

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
