# Peer review of "Photovoltaic Fuzzy Logical Control MPPT Based on Adaptive Genetic Simulated Annealing Algorithm-Optimized BP Neural Network"

_processes, doi:10.3390/pr10071411_

Round 1

Reviewer 1 Report

There are major concerns that must be re-considered to comprehensively improve the content to re-discuss the possibility of accepting a paper study. The reviewer hopes the authors will find the below-given comments useful to improve their study:

-        The Turnitin similarity rate of the article is as high as 27%, so the paper should recheck.

-        Sections of the paper should be reorganized and rearranged.

The information (3.1 and 3.2) given in the third section under the "Related Works" heading should be included in the second section as 2.2 and 2.3 after the subheading 2.1. In the third part, existing sub-headings 2.2 and 3.3 should be included as 3.1 and 3.2. In other words, information in the form of a literature summary should be given under a separate title, and information on the study should be conducted under another title.

-        The proposed model in the paper was analyzed only with the simulation made in the MATLAB program. At least whether real-time tests will be carried out with the physical circuit design and application of the proposed model under the title of the future work should be mentioned.

-        Publication names given as abbreviations in the References should be checked and rearranged. For example, “Gener Transm Dis.”, “Knowl-Based Syst.”, “Renew Energ.”,  and “Appl Soft Comput.”

All the sources should be checked and written in scientific reference style.

Author Response

Dear Review 1,

Best wishes,

Yan Zhang

Reviewer 2 Report

The authors applied an adaptive GA is adopted to generate the corresponding population and increase the population diversity. The simulated annealing (SA) algorithm is applied to the parent and offspring with higher fitness value to improve the convergence rate of GA, and the optimal weight threshold of BPNN are updated by GA and SA algorithm. Then, the optimized BPNN is employed to predict the MPP voltage of PV cell. Finally, the fuzzy logical control (FLC) was used to eliminate local power oscillation and improve the robustness of PV system. The proposed method was employed and compared to GA-BPNN, simulated annealing-genetic (SA-GA) and FLC algorithm under the condition that both the irradiation and temperature change.

The paper is well presented and structured. The developed method is also nice and the results are good. It can be accepted for publication, with some changes, such as:

- First, you have to give a reason why you select GA and SA. There are more recent optimization algorithms that published in recent years, so, why you select those old methods?

- Second, what are the criteria you used to select the compared methods? And how did you guarantee fair comparisons?

- Also, why you did not consider other algorithms for comparisons, or discussions?

- There are other methods, that can be used for discussions such as Random reselection particle swarm optimization for optimal design of solar photovoltaic modules; Modified aquila optimizer for forecasting oil production; Increasing electric vehicle autonomy using a photovoltaic system controlled by particle swarm optimization; Parameter extraction of photovoltaic module using tunicate swarm algorithm;

- Improve the quality of some figures.

- Some captions are not informative, improve them.

-Improve the conclusion section. Support it by numerical results.

Author Response

Dear Reviewer 2,

Best wihes,

Yan Zhang

Round 2

Reviewer 1 Report

The paper is better organized and explains the Photovoltaic Fuzzy Control MPPT using Adaptive Genetic Simulated Annealing Algorithm-Optimized BP Neural Network with details after the major revision. Reviewer thanks to authors for their comprehensive and well-written scientific paper.

Author Response

Dear reviewer,

Best wishes,

Yan Zhang

Reviewer 2 Report

The authors did not take our comments seriously. You have to do each comment one by one:

-First, English needs more modifications, for example, gray wolf is >>grey wolf

- In some figures you put some symbols without defining their meaning, for example, figure 5, what are the w, f, v, and others mean? You have to explain each symbol in the figures.

- In figure 10, (best and goal) lines are not clear, do not use similar color. Use another one.

- Give more details about the parameters you mentioned in Table 3. Why you did not use other settings.

- Improve the presentation of Figure 11. Also, please not this figure has no caption. You have to be serious and to check everything in your paper.

- There are other methods, that can be used for discussions such as Random reselection particle swarm optimization for optimal design of solar photovoltaic modules; Modified aquila optimizer for forecasting oil production; Increasing electric vehicle autonomy using a photovoltaic system controlled by particle swarm optimization; Parameter extraction of photovoltaic module using tunicate swarm algorithm;

- Statistical tests are needed to examine the quality of the methods. And to show the differences between the compared methods. So, you may apply Friedman test , Wilcoxon test, and p-value.

- Deeply discuss the limitations. 

- Also remove the " research on " from the title, no need for it. 

Author Response

Dear reviewer,

Best wishes,

Yan Zhang

Round 3

Reviewer 2 Report

The authors have addressed all comments. This version can be accepted.